# Elliptical Space with the McVittie Metrics

Vladimir N. Yershov [†]

Institute of Radio-Engineering and Infocommunication Technology, State University of Aerospace Instrumentation, 67 Bol'shaya Morskaya, 190000 St Petersburg, Russia; vyershov@moniteye.co.uk

[†] Former address: Mullard Space Science Laboratory, UCL Department of Space and Climate Physics, Holmbury St. Mary, Dorking RH5 6NT, UK.

**Abstract:** The main feature of elliptical space—the topological identification of its antipodal points—could be fundamental for understanding the nature of the cosmological redshift. The physical interpretation of the mathematical (topological) structure of elliptical space is made by using physical connections in the form of Einstein-Rosen bridges (also called "wormholes"). The Schwarzschild metric of these structures embedded into a dynamic (expanding) spacetime corresponds to McVittie's solution of Einstein's field equations. The cosmological redshift of spectral lines of remote sources in this metric is a combination of gravitational redshift and the time-dependent scale factor of the Friedmann-Lemaitre-Robertson-Walker metric. I compare calculated distance moduli of type-Ia supernovae, which are commonly regarded as "standard candles" in cosmology, with the observational data published in the catalogue "Pantheon+". The constraint based on these accurate data gives a much smaller expansion rate of the Universe than is currently assumed by modern cosmology, the major part of the cosmological redshift being gravitational by its nature. The estimated age of the Universe within the discussed model is $1.48 \cdot 10^{12}$ yr, which is more than two orders of magnitude larger than the age assumed by using the standard cosmological model parameters.

**Keywords:** elliptical space; de Sitter metric; Schwartzschild metric; McVittie metric; wormholes; gravitational redshift; type-Ia supernovae

## 1. Introduction

Hamilton (1843) discovered quaternions [1] which represent rotational geometry related to elliptical space (also called projective space, $\mathbb{P}^3$). This space was introduced to cosmology by de Sitter [2] to replace the hyper-spherical space $\mathbb{S}^3$ of Einstein's first cosmological model [3]. De Sitter argued that elliptical space is preferable for modelling the physical world, rather than $\mathbb{S}^3$. This was also the opinion of Einstein communicated to de Sitter by letter. The main argument was based on the observation that when used for projecting natural coordinates in Euclidean (flat) or Lobachevsky (hyperbolic) space, a sphere $\mathbb{S}^3$ covers them twice, which is ambiguous. However, the elliptical space $\mathbb{P}^3$ covers them only once, which avoids the ambiguity. This property of elliptical space follows from its main feature: the topological identification of its antipodal points corresponding to the projective angle $\chi = \pi$.

Some researchers argue that the notion of elliptical space is outdated, and that it is more familiar as "the de Sitter spacetime in static coordinates" because de Sitter used it for exploring a static variant of his cosmological model. But the main point is not the use of static or dynamic coordinates in elliptical space (both can be used). The importance of elliptical space is in the topological identification of its antipodal points. Elliptical space was previously explored by Newcomb in 1877 [4] and Schwarzschild in 1900 [5]. Lemaître [6], Tolman [7] and Robertson [8–10] explicitly used the term "elliptical space" in their expanding-universe models, including the model with the Friedmann-Lemaître-Robertson-Walker metric (FLRW). From the latter model, elliptical space tacitly evolved into the modern ΛCDM model, although some researchers might be unaware of it.

Elliptical space can be embued with any metric satisfying Einstein's field equations of general relativity. The Einstein metric

$$ds^2 = c^2 dt^2 - dr'^2 - R^2 \sin^2 \frac{r'}{R} (d\theta^2 + \sin^2 \theta d\varphi^2) \tag{1}$$

of a static Riemannian space with constant positive curvature

$$\lambda = R^{-2}, \tag{2}$$

was the first of such metrics. It corresponds to a matter density $\rho > 0$, with time of the whole Universe given by the unit time-related metric coefficient in (1). De Sitter found yet another metric satisfying the Einstein field equations:

$$ds^2 = \cos^2 \frac{r'}{R} c^2 dt^2 - dr'^2 - R^2 \sin^2 \frac{r'}{R} (d\theta^2 + \sin^2 \theta d\varphi^2). \tag{3}$$

In this model, $\lambda = 3R^{-2}$ but $\rho_0 = 0$, which implies the absence of matter and, hence, the absence of observers in such a universe. The time-related metric coefficient ($g_{tt}$) here varies with distance from the observer. So, time is not universal anymore because $g_{tt} = \cos^2 \frac{r'}{R} = \cos^2 \chi$ decays to zero at the maximal distance from the observer corresponding to $\chi = \pi/2$, which implies the complete cessation of time.

In Equations (1) and (3), $r'$ is the natural radial coordinate as measured by the projective angle $\chi$:

$$r' = R\chi, \tag{4}$$

with the origin of coordinates at the observer's location. This is schematically illustrated by Figure 1 where the main feature of elliptical space—the topological identification of antipodal points—is indicated by the vertical dashed line down from the observer's location on the sphere $\mathbb{S}^3$. De Sitter introduced the coordinate transformation

$$r = R \tan \chi, \tag{5}$$

in which $r$ is the projection of the natural coordinate $r'$ onto Euclidean or Lobachevsky space tangential to $\mathbb{S}^3$ at the observer's location $o$.

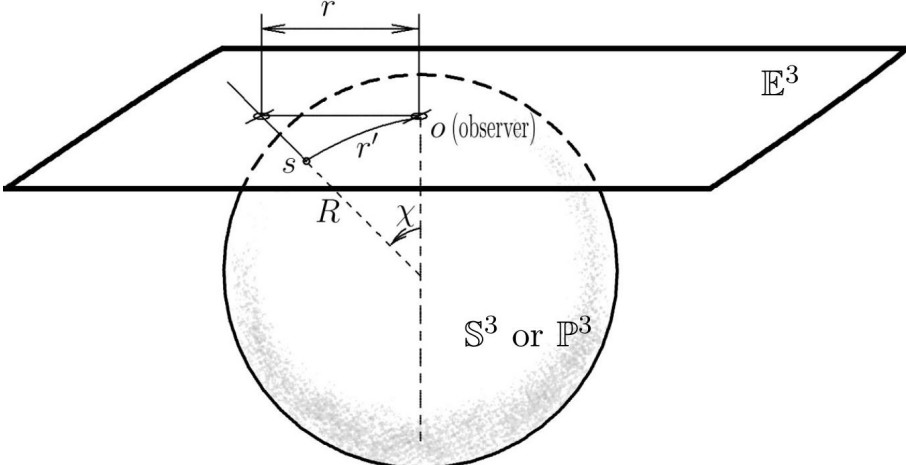

**Figure 1.** Embedding diagram (one spatial dimension suppressed) depicting spherical $\mathbb{S}^3$ or elliptical $\mathbb{P}^3$ space of constant positive curvature with two antipodal points topologically identified (the vertical dashed connection between two poles). The tangential Euclidean space $\mathbb{E}^3$ at the observer's location $o$ indicates the local coordinate reference frame. Distances $r'$ along the natural spatial coordinate of $\mathbb{S}^3$ are measured by the projective angle $\chi$, with $r' = R\chi$. The corresponding projective distance to the source $s$ in $\mathbb{E}^3$ is $r = R \tan \chi$.

In the de Sitter's solution, there is a global redshift effect $z_{dS}$ (called the de Sitter effect) caused by time dilatation due to the metric coefficient

$$g_{tt} = \cos^2 \chi \qquad (6)$$

in the de Sitter metric (3). According to the definition $1 + z_{dS} = g_{tt}^{-\frac{1}{2}}$, the de Sitter effect is the *cosmological gravitational redshift* for a homogeneously distributed energy density $\rho_\lambda = \lambda c^2/(8\pi G)$. Unlike the local gravitational redshift, which is always anisotropic from the observer's perspective, the time dilatation due to the de Sitter effect is spherically symmetric around any arbitrary point of space (i.e., it is isotropic). This was de Sitter's prediction of the cosmological redshift phenomenon [2] made a decade before its observational discovery in 1927 by Lemaître [6] and in 1929 by Hubble [11].

In the standard $\Lambda$CDM cosmological model, the cosmological redshift is interpreted in terms of the motion of recession within the expanding space paradigm encoded by the Robertson-Walker's time-dependent cosmic scale factor $a(t)$[1] of the FLRW metric [6,8,12,13]. The theoretical Hubble diagram based on the FLRW metric fits almost perfectly the observational data collected in the form of distance moduli of 1701 type-Ia supernovae in the Pantheon+ catalogue [14,15]. The goodness-of-fit parameter ($\chi^2$) of the $\Lambda$CDM model to these data is pretty small: $\chi^2_{\Lambda CDM} = 901.6$, the parameters $\Omega_m = 0.334$ and $H_0 = 73.6 \, \text{km s}^{-1} \, \text{Mpc}^{-1}$, $\Omega_k = 0$, $\Omega_\Lambda = 1 - \Omega_m$, where this fit has been taken from [14].

By contrast, the $\chi^2$ computed for de Sitter's model is very large (orders of magnitude), which can be explained by the fact that, in this model, the redshift-distance relationship is non-linear (quadratic) for small redshifts, whereas the observational redshift-distance relationship is strictly linear. That is why the de Sitter concept of global gravitational redshift in a static universe was abandoned, while the expanding-universe paradigm prevailed because the $\Lambda$CDM model based on this paradigm successfully predicted and explained many observational phenomena, including the abundances of light elements in the Universe due to the Big-Bang nucleosynthesis, the isotropy and the power-spectrum of the Cosmic Microwave Background Radiation (CMBR), the power-spectrum of matter overdensities, and more.

However, despite the successes of the standard cosmological model during the last ninety years, there exist numerous observational facts casting doubt on the validity of this model. For example, the perturbative approach used in $\Lambda$CDM to explain the structure-formation in the Universe is challenged by the fact that on scales below 100 Mpc the matter distribution in the Universe is extremely inhomogeneous, which is also related to some other issues with $\Lambda$CDM [16]. Statistical studies of matter distribution [17–19] and of the CMBR [20] indicate that the Universe is fractal, which is in line with earlier theoretical studies of static cosmological models [21,22]. According to the $\Lambda$CDM scenario, fractal distribution of matter is impossible. The more so, because of the difficulties of explaining within the $\Lambda$CDM framework the origin of the largest-scale structures, such as a huge arc spanning 1 Gpc at redshift $z \sim 0.8$ [23], large filaments or wall-like super-clusters with huge voids between them [24].

Other problems are related to the standard Big-Bang nucleosynthesis theory [25–27] and to the discrepancy (tension) between the values of the Hubble constant measured by different methods [28,29]. With respect to the latter issue, the present author found that the discrepancy is likely related to the fact that the CMBR data obtained by the Wilkinson Microwave Anisotropy Probe (WMAP) were contaminated by the irreducible intergalactic foreground [30], which was later confirmed by the *Planck* mission data [31,32]. Also consistent with those studies are findings of the CMBR hot and cold spots correlation with matter overdensities and underdensities [33,34]. The famous Cold Spot is notoriously inexplicable within the $\Lambda$CDM framework. However, it was found to be physically related to the Eridanus supervoid [35,36], which links the CMBR origin with the matter distribution in the Universe.

The most recent and the most problematic issue in the standard cosmological model came from the James Webb Space Telescope (JWST) observations [37,38]. This telescope, capable of capturing images of galaxies with redshifts well above $z = 10$, discovered numerous high-redshift galaxies, which, by their properties turned out to be very similar to the galaxies in the local universe. These remote galaxies were found to be fully developed, despite having no time for their development from the beginning of the Universe as calculated within the ΛCDM framework. The observed number densities of massive galaxies with redshifts above $z = 10$ are also inconsistent with the galaxy formation models based on the ΛCDM predictions [39].

These new observations indicate that the standard model's prediction with respect to the age of the Universe is largely incorrect [40]. This dilemma can be solved by cosmological models that provide more time for high-redshift galaxies to evolve. There are recent publications discussing the possibility of increasing the estimated age of the Universe, either hypothesising the variability of fundamental constants [41] or by incorporating a "zero active mass" [42]. In this paper I discuss a similar possibility, that of increasing the Universe's age, but without non-physical assumptions. My model uses de Sitter's formulation with a global static gravitational redshift combined with the dynamic redshift based on a FLRW-like metric. The static part of the global redshift based on the Schwarzschild metric has already been discussed by the author elsewhere [43]. Here I resort to the combination of the Schwarzschild and FLRW metrics found in 1933 by G. C. McVittie [44]. In order to devise a strictly linear distance-to-redshift relationship, I make use of the main feature of elliptical space—the identification of its antipodal points, which is described in the next section.

## 2. Materials and Methods

### 2.1. The Origin of Coordinates

The global de Sitter effect [2] is due to the difference between the local (observer's) and remote source coordinate reference frames. Another global gravitational redshift effect was discovered in 1947 by H. Bondi [45] who considered the Einstein metric (1) and found that there is a redshift proportional to the gravitational potential difference between the surface and the center of a ball of matter centred at the source and having radius equal to the source-to-observer distance. In both de Sitter and Bondi's cases, the redshift-distance dependence is non-linear for small redshifts, which is at odds with observations.

In order to devise the required strictly-linear relationship between the source's redshift and its distance to the observer, I resort to an unusual method of translating the origin of coordinates from the observer's location to the observer's antipodal point (to the best of the author's knowledge, in all previous cosmological models, the origin of coordinates is at the observer's location). For this method to work, one needs to endue the observer's antipodal point with the Schwarzschild metric, which will provide the gravitational redshift effect.

One can achieve this by interpreting the main mathematical property of elliptical space—the identification of its antipodal points—as a physical, direct connection between these points. In terms of physics, the direct connection between remotely separated points of space is described by a general-relativistic structure found in 1935 by A. Einstein and N. Rosen [46]. This structure is called the Einstein-Rosen bridge or, more frequently, a "wormhole". It has two "throats", one in the vicinity of the observer (a near throat), and another at a very large distance from the observer (the far-throat). The wormhole connects two different spaces or two remote locations of the same space. Both of its throats are endued with the Schwarzschild metric, which can be used for calculating the gravitational redshift effect. In the case of elliptical space, the connection is between two antipodal points. But, in principle, other possible connections are not excluded.

Some authors argue that short-circuited connections between remotely separated points of space via wormholes explain the phenomenon of quantum entanglement of subatomic particles separated by large distances [47]. Wormholes were also considered by

Morris, Thorne & Yurtsever as structures that allow instantaneous travel between remote regions of space [48]. However, the same authors, while further studying this possibility in detail, found that wormhole creation requires extremely large energies [49], which in terms of particle physics corresponds to extremely small distances. Thus, wormholes are likely to exist only on a scale-length of the order of the Planck-Wheeler length, $\ell_P = \sqrt{G\hbar/c^3} = 1.62 \cdot 10^{-33}$ cm. So, they are microscopic particles. There is a belief based on some past research (for example [50]) that wormholes are unstable and would instantly collapse once anything, such as a photon, traversed the wormholes' horizon. But microscopic wormholes are not traversable, as they are the smallest possible entities of the Planck-Wheeler length with no singularities. This is also concurs with M.A. Markov's suggestion [51] that the density of matter is always subject to an upper limit given by the expression

$$\rho_q = \frac{c^5}{G^2\hbar}. \tag{7}$$

This follows logically because infinite energies are never observed in the Universe. As in the case of paper [50], practically the whole of the wormhole stability research is related to either wormhole traversability by matter or to the wormhole's matter of field content (see [52–54] and others). The methods which are used include modified or extended theories of gravity [55], or quantum gravity [56]. None of these studies correspond to the ideas of Einstein, Rosen and Wheeler which posed the question about the origin of matter. Their microscopic wormholes are the smallest matter particles or matter particle ingredients. By definition they are stable and not traversable.

In elliptical space, the observer's antipodal point is seen from the observer's perspective as a sphere with a very large radius around the observer. So, what is seen as a point near the observer, is also seen as a global sphere spanning the whole $4\pi$ steradians around. This is schematically illustrated by a diagram in Figure 2.

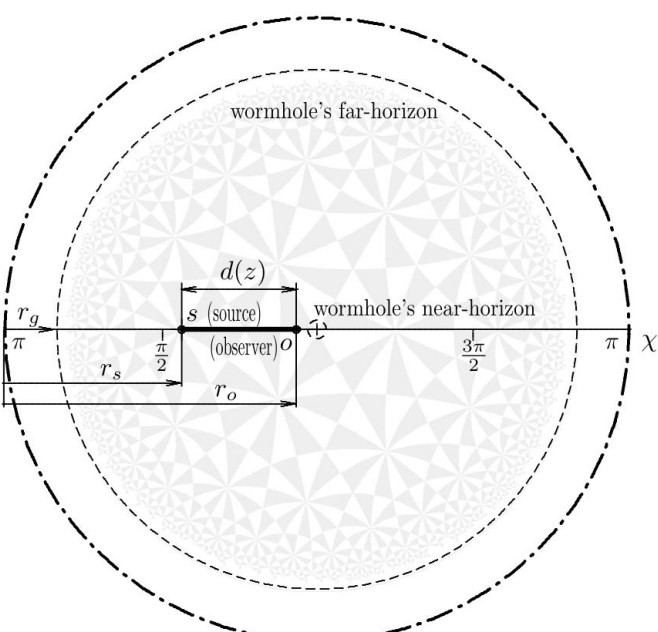

**Figure 2.** Poincaré disk illustrating the concept of the spherical symmetry of wormhole far-throats around each point in elliptical space. The disk spans the whole of the $\mathbb{S}^3$ sphere, the central point of the disk (the centre of the wormhole's near-throat) being near the observer's location $o$. A small dashed circle around the central point denotes the event horizon of the near-throat, and the larger dashed circle denotes the corresponding event horizon of the wormhole's far throat. The distances $r_s$, $r_o$ and $r_g$ are measured from the centre of the far-throat (the outer dot-dashed circle).

This diagram is a Poincaré disk conformally mapping the whole space to a (unit) disk with its circumference (the above mentioned global sphere) drawn in the form of a dot-dashed circle representing the antipodal point of the disk's centre. The disk's centre and its antipodal point (the circumference of the disk) are topologically identified as the same entity, i.e., they are physically connected via a wormhole.

A small dashed circle around the center of the Poincaré disk depicts the wormhole's near-throat (a sphere) with its microscopic Schwartzschild radius. This small sphere corresponds to a large antipodal sphere at some distance from the disk's circumference towards the interior of the disk. In Figure 2, it is shown as a large dashed circle at some distance $r_g$ from the disk edge. Although the observer is nearby and *outside* the small sphere, yet it is *inside* it because the antipodal image of this sphere surrounds the observer at a large distance.

### 2.2. Isotropy and Spherical Symmetry

A large volume of space around the observer, which includes the observer and all surrounding sources of light (the Sun, stars, galaxies, clusters of galaxies) is seen from the observer's perspective as a very distant layer of space adjacent to the wormhole's far-horizon. An essential aspect of this construct is its spherical symmetry for any arbitrary location in space. Assuming that the Universe is homogeneous on a large scale, one can show that for a sufficiently large neighbourhood around the observer all sources located within this volume are seen as a uniformly illuminated spherical layer bordering on the far-horizon. This layer is at a very large distance—much farther away from the observer than the limit of the observer's neighbourhood. A possible mathematical description of a similar topological boundary around a point in a finite-volume set was conceived for both hyperbolic and elliptic geometries [57,58].

The remote horizons corresponding to each point within the neighbourhood region are at extremely large distances from the observer. This region might appear large, as it likely to include many clusters of galaxies. But cosmologically, its size is negligible as compared with the distances to the remote horizons of the points belonging to this region. In this way, a remote, almost spherically symmetric, collective horizon is formed around each point of elliptical space, thus, making any observer's location equivalent to any other location. This conforms to the cosmological principle of isotropy.

### 2.3. Schwarzschild Metric

As just mentioned above, a sphere of neighbourhood matter with mass $M$ surrounding the observer produces the effect of a global collective remote horizon, which is due to the Schwarzschild metric of the collective far-horizons of wormholes in the observer's neighbourhood. The corresponding spacetime interval in Schwarzschild coordinates is [59]:

$$ds^2 = g_{tt}c^2dt^2 - g_{rr}dr^2 - r^2d\Omega^2, \tag{8}$$

where $d\Omega = d\theta^2 + \sin^2\theta d\varphi^2$. Here the metric coefficients $g_{tt}$ and $g_{rr}$ are

$$g_{tt} = 1 - \frac{2GM}{c^2 r} \quad \text{and} \quad g_{rr} = g_{tt}^{-1}, \tag{9}$$

where $\frac{2GM}{c^2} = r_g$ is the gravitational (Schwarzschild) radius. This metric can also be expressed in isotropic coordinates as

$$ds^2 = \left(\frac{1 - \frac{GM}{2c^2 r}}{1 + \frac{GM}{2c^2 r}}\right)^2 c^2dt^2 - \left(1 + \frac{GM}{2c^2 r}\right)^4 (dr^2 + r^2d\Omega^2) \tag{10}$$

or

$$ds^2 = \left(\frac{1 - \frac{r_g}{4r}}{1 + \frac{r_g}{4r}}\right)^2 c^2dt^2 - \left(1 + \frac{r_g}{4r}\right)^4 (dr^2 + r^2d\Omega^2). \tag{11}$$

We shall use this form when comparing this metric with the dynamic case. In the above formulae, $r$, $\varphi$ and $\theta$ are spherical coordinates, $G$ is the gravitational constant, $c$ is the speed of light, and $M$ is the mass of a very large sphere of matter surrounding the observer and producing the effect of a global remote horizon with its corresponding global gravitational redshift effect. As already mentioned, this effect is spherically symmetric for an arbitrary point of space. The mass $M$ and the size of the sphere of matter are determined by the causal connectivity corresponding to the finite speed of the gravitational interaction.

In accord with our choice of the origin of coordinates at the observer's antipodal point, any distance is measured from this antipodal point (which is a large sphere around the observer) towards the source ($r_s$) and towards the observer ($r_o$). The gravitational radius $r_g$ is also measured from this sphere. These distances are indicated in Figure 2 by the pointers at the left edge of the sphere. Our purpose here is to find a relationship between the source redshift ($z$) and the source-to-observer distance ($d$):

$$d(z) = r_o - r_s \,. \tag{12}$$

Both $r_o$ and $r_s$ distances are unknown, but the latter can be expressed in terms of the distance $d(z)$. The parameter $r_g$ is also to be determined by using observational data. In the simplest Schwarzschild case, there are two free parameters: $r_o$ and $r_g$. Both source and observer are located within the Schwartzschild metric (8) with its redshift-defining coefficient (9). So, the source's redshift with respect to the observer is

$$z = \sqrt{\frac{g_{tt}^o}{g_{tt}^s}} - 1 \tag{13}$$

or, by taking into account (9),

$$(z+1)^2 = \frac{1 - \frac{r_g}{r_o}}{1 - \frac{r_g}{r_s}} \,. \tag{14}$$

For simplicity, here we shall assume the parameter $r_g$ to be our distance unit. That is, we put $r_g = 1$. Later on, it can be converted to some common distance unit, such as Mpc. Thus,

$$(1+z)^2 = \left(1 - r_o^{-1}\right)\left(1 - r_s^{-1}\right)^{-1}. \tag{15}$$

By using (12) we replace $r_s$ by distance $d$:

$$r_s = r_o - d \,, \tag{16}$$

obtaining

$$(r_o - d)^{-1} = 1 - (1 - r_o^{-1})(1+z)^{-2} \tag{17}$$

and, finally,

$$d = r_o - \left[1 - (1 - r_o^{-1})(1+z)^{-2}\right]^{-1}, \tag{18}$$

which is the required expression [in units of $r_g$] for calculating the theoretical source-to-observer distance for a given source redshift. For this redshift, distance $d$ can be compared with the observed luminosity distance of a source. For this comparison, (18) has to be converted to the luminosity distance by using the squared $(1+z)$-factor:

$$d_L(z) = d(1+z)^2 \,. \tag{19}$$

This factor is formed of three sub-factors accounting

- $(1+z)^{\frac{1}{2}}$ —for the loss of luminosity due to the cosmological redshift $z$;
- $(1+z)^{\frac{1}{2}}$—for the lower rate at which the photons reach the observer because of the cosmological time dilatation;

- $(1+z)$— for the photon path distortion (the $g_{rr}$ coefficient of the metric).

  If the distance unit $r_g$ is expressed in Mpc, then the theoretical distance modulus

$$\mu = 5\log(r_g d_L) + 25 \tag{20}$$

is comparable with the observed distance moduli of type Ia supernovae (in stellar magnitudes).

*2.4. McVittie Metric*

The McVittie metric [44] describes the gravitational field of a mass point embedded into a dynamic FLRW metric with the scale factor

$$a(t) = \mathrm{e}^{Ht}, \tag{21}$$

where $H$ is a Hubble-like constant. It also includes the space curvature $k$, which I omit for simplicity. For the flat-universe case with $k = 0$, the McVittie metric in isotropic coordinates is

$$ds^2 = \frac{\left(1 - \frac{GM}{2c^2 ra(t)}\right)^2}{\left(1 + \frac{GM}{2c^2 ra(t)}\right)^2} c^2 dt^2 - a(t)^2 \left(1 + \frac{GM}{2c^2 ra(t)}\right)^4 (dr^2 + r^2 d\Omega^2). \tag{22}$$

For the mass parameter $M = 0$, the McVittie metric coincides with the FLRW metric, and for the constant $a(t) = 1$, it is reduced to the Schwarzschild metric (10) in isotropic coordinates. By putting

$$\tilde{r}_g = \frac{r_g}{a(t)} = \frac{2GM}{c^2 a(t)} \tag{23}$$

one gets the above expression in the form

$$ds^2 = \frac{\left(1 - \frac{\tilde{r}_g}{4r}\right)^2}{\left(1 + \frac{\tilde{r}_g}{4r}\right)^2} c^2 dt^2 - a(t)^2 \left(1 + \frac{\tilde{r}_g}{4r}\right)^4 (dr^2 + r^2 d\Omega^2), \tag{24}$$

which has the same form as Equation (11). Therefore, the luminosity distance (18) and (19), which is given in units of $r_g$, can be used for the McVittie metric case, if one takes care of the time-dependent distance unit (23).

The time $t$ in (21) can be calculated, to the first approximation, by using the static distance $d_L$ divided by the speed of light ($c = 1$ in units of $r_g$) with the first correcting sub-factor in (19) removed, because the loss of luminosity due to the cosmological redshift $z$ is not applied when calculating the light-travel distance. That is,

$$t = d(1+z)^{\frac{3}{2}}. \tag{25}$$

Then Equation (20) can be re-written as

$$\mu = 5\log(\tilde{r}_g d_L) + 25. \tag{26}$$

Finally, the theoretical model for fitting the observed type-Ia supernova distance moduli consists of Equations (18), (19), (25), (21), (23) and (26) with three free parameters: $r_o$, $H$ and $r_g$, the first two expressed in units of $r_g$ and the third parameter in Mpc.

## 3. Results

*3.1. Parameter Estimation*

By applying the formulae derived above for the McVittie metric, one can calculate the theoretical luminosity distances $d_L^{\mathrm{McV}}$ and distance moduli $\mu^{\mathrm{McV}}$ corresponding to the redshifts of the type-Ia supernova. The comparison of these distance moduli with the observational data from the Pantheon+ catalogue is presented in Figure 3. The fit

to the observational data is implemented by minimising the Pearson $\chi^2$ criterion for three parameters: $r_o$, $H$ and $r_g$. The minimum with $\chi^2_{\text{McV}} = 886.8$ corresponds to the following parameter values:

$$r_o - 1 = (9.744^{+0.016}_{-0.015}) \cdot 10^{-8}, \tag{27}$$

$$H = (4.770 \pm 0.063) \cdot 10^4, \tag{28}$$

$$r_g = (2.160 \pm 0.003) \cdot 10^{10} \, [\text{Mpc}], \tag{29}$$

the parameter $r_o$ being in units of $r_g$, the Hubble-like parameter $H$ in units of time of light travel across $r_g$. The parameter $r_g$ is expressed in Mpc by the choice of coefficients in (26).

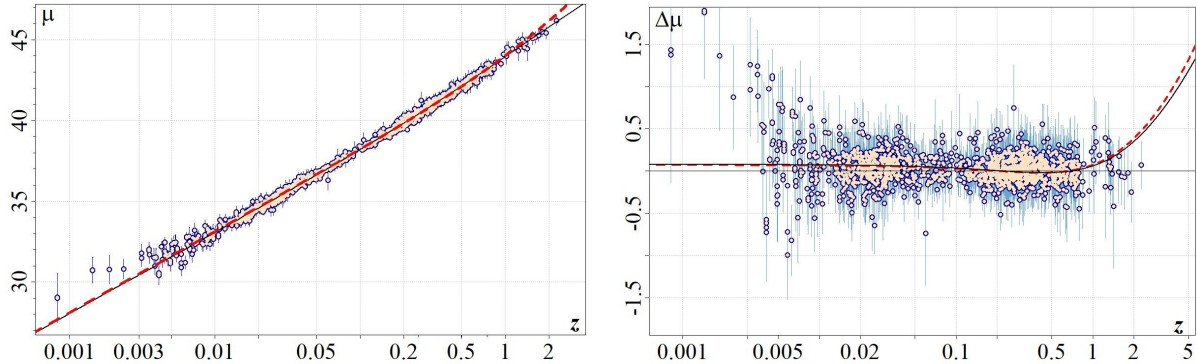

**Figure 3.** Left: comparison of theoretical distance moduli $\mu$ (the red and black curves) with the observational distance moduli of 1701 type-Ia supernovae from the Pantheon+ catalogue (the salmon-colour points). The dashed red curve corresponds to the model based on the McVittie metric, while the solid black curve is the reference corresponding to the standard $\Lambda$CDM cosmological model. Right: the same data in detail, plotted in the form of the residuals $\Delta\mu$ with respect to the $\Lambda$CDM-reference, which is thus the horizontal line at $\Delta\mu = 0$. The abscissae in both plots correspond to source redshifts $z$.

The salmon-colour points in Figure 3 represent the observational data from Pantheon+ [14]. The McVittie-based theoretical distance moduli for the minimum of $\chi^2_{\text{McV}}$ are plotted in the form of a red dashed curve, and the $\Lambda$CDM fit from [14] is shown as the black curve. The right panel of Figure 3 shows the differential data points $\Delta\mu$ with respect to the $\Lambda$CDM theory (the thin horizontal line at $\Delta\mu = 0$). The thick black curve on the right panel corresponds to the McVittie-based model with the parameter $H = 0$. By comparing the previously mentioned goodness-of-fit parameter $\chi^2_{\Lambda\text{CDM}} = 901.6$ with $\chi^2_{\text{McV}} = 886.7$, one can see that the latter compares with observational data to the same level of accuracy as $\Lambda$CDM. That is, both models are equivalent in terms of fitting their theoretical Hubble diagrams to observations within the redshift range covered by the Pantheon+ data ($z < 2.4$). However, for higher redshifts these models diverge. This is shown on the left panel of Figure 4, where the theoretical luminosity distances $d_L$ are plotted for the $\Lambda$CDM model (the black curve) with the parameters $\Omega_m = 0.334$ and $H_0 = 73.6 \, \text{km s}^{-1} \, \text{Mpc}^{-1}$, $\Omega_k = 0$, $\Omega_\Lambda = 1 - \Omega_m$ from [14] and for the model based on the McVittie metric discussed here (the red curve) for the parameters (27)–(29). The black dashed curve is shown by comparison with the static version of the latter model, i.e., when $H = 0$.

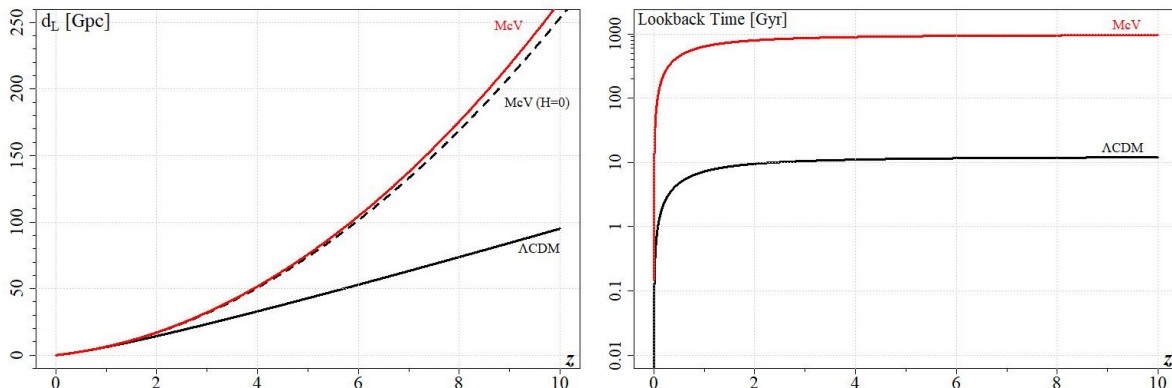

**Figure 4.** Left: theoretical luminosity distances (in Gpc) as predicted by the $\Lambda$CDM model (the black curve), the model based on the McVittie metric (McV, the red curve), and the McV model with the Hubble-like parameter $H = 0$ (the dashed black curve). Right: look-back time for the $\Lambda$CDM model with the Hubble parameter $H_0 = 73.6 \frac{\text{km/s}}{\text{Mpc}}$ (black curve) and for the model based on the McVittie metric with its Hubble-like parameter $H = 0.6621 \frac{\text{km/s}}{\text{Mpc}}$ (red curve).

*3.2. Age of the Universe*

Knowing the value (29) of the parameter $r_g$, one can estimate the corresponding unit of time, $t_u$, which is the light-crossing time of the distance $r_g$ (the speed of light, $c = 1$, is also expressed in units of $r_g$). With $1\,\text{Mpc} = 3.0857 \cdot 10^{22}\,[\text{m}]$ and $c = 2.998 \cdot 10^8\,[\text{m/s}]$

$$t_u = \frac{r_g \times 3.0857 \cdot 10^{22}\,[\text{m}]}{2.998 \cdot 10^8\,[\text{m/s}]} = 2.223 \cdot 10^{24}\,[\text{s}], \tag{30}$$

the Hubble-like parameter $H$ expressed in metric units reads

$$\frac{H \times r_g[\text{m}]}{t_u[\text{s}] \times r_g[\text{Mpc}]} = \frac{4.77 \cdot 10^4 \times 6.665 \cdot 10^{32}}{2.223 \cdot 10^{24} \times 2.16 \cdot 10^{10}} = 662 \left[\frac{\text{m/s}}{\text{Mpc}}\right]. \tag{31}$$

Thus $H = 0.662 \pm 0.009 \frac{\text{km/s}}{\text{Mpc}}$, assuming the tolerance intervals from (28). The right panel of Figure 4 contrasts the look-back time based on this value (the red curve) with the look-back time based on the $\Lambda$CDM model ($H_0 = 73.6 \frac{\text{km/s}}{\text{Mpc}}$, the black curve). The parameter $H$ does not vary with time, as can be seen from (21):

$$\frac{\dot{a}}{a} = H = \text{const.} \tag{32}$$

In terms of frequencies, $H = 2.146 \cdot 10^{-20}\,[\text{s}^{-1}]$, which constrains the age of the Universe to $H^{-1} = 4.66 \cdot 10^{19}\,[\text{s}]$ or

$$H^{-1} = (1.48 \pm 0.02) \cdot 10^{12}\,[\text{yr}], \tag{33}$$

assuming the tolerance intervals from (28).

## 4. Discussion

The cosmological model discussed here gives a new interpretation of the redshift-luminosity distance relationship for type-Ia supernovae. In this interpretation, the cosmological redshift is based on the static Schwarzschild metric embedded in the expanding FLRW metric, so the major part of the cosmological redshift is gravitational by its nature.

Therefore, the Hubble-like constant (31) is very small, and the corresponding age of the Universe (33) based on this constant is extremely large. The look-back time calculated by using the parameter (31) for $0 < z < 10$ with $\Omega_m = 1, \Omega_\Lambda = 0$ is plotted on the right panel

of Figure 4 (the red curve). It is about two orders of magnitude larger than the commonly accepted value.

By assuming the matter density $\rho_m \propto a(t)^{-3}$ and the radiation density $\rho_{rad} \propto a(t)^{-4}$ one can evaluate their ratio $\rho_m/\rho_{rad}$ at the end of the radiation-dominated era, when $\rho_m = \rho_{rad}$ (see Figure 5).

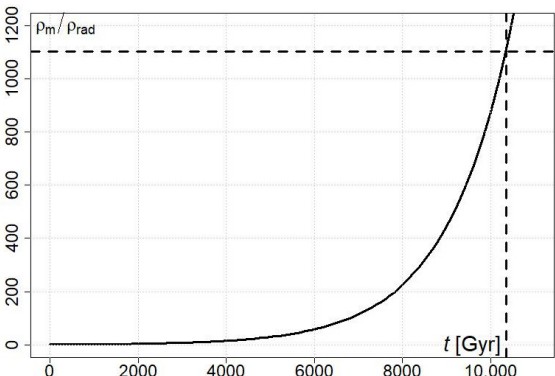

**Figure 5.** Matter to radiation density ratio from the end of the radiation-dominated era, as calculated from the model based on the McVittie metric by using the parameter (31).

The observational present-day ratio $\rho_m/\rho_{rad} \approx 10^3$ is indicated in Figure 5 by the horizontal dashed line. It corresponds to the time $\approx 10^{13}$ [yr] (the vertical dashed line). Presumably, this time is required for reaching the present-day matter-to-radiation density ratio. But it is at odds with the age of the Universe (33) previously estimated in Section 3.2.

The most likely reason for this inconsistency is that using the Friedmann equations for deriving the evolution of matter and radiation density in the discussed model is not fully adequate for achieving the observed ratio of these quantities. Further investigation of this question is required. This inconsistency, together with the smallness of the Hubble-like parameter $H = 0.662 \left[\frac{\text{km/s}}{\text{Mpc}}\right]$ of the McVittie metric, makes the discussed model practically indistinguishable from static. Therefore, here we have to pay regard to the observational challenges of a static cosmological model.

The standard cosmological model, $\Lambda$CDM, based entirely on the dynamical interpretation of the cosmological redshift due to the recession velocities of galaxies, is very successful in explaining many other observational facts. Therefore, universal opinion tends to consider $\Lambda$CDM as the best available model, while any alternative models, especially those based on the static (or almost static) metric must demonstrate that they can be, at least, as successful as $\Lambda$CDM in explaining not only the redshift-luminosity distance relationship (discussed here), but all other observational facts, plus dark matter and dark energy, which are two essential components of $\Lambda$CDM.

### 4.1. Dark Energy

The comparison of the theoretical distance moduli with observations (see Figure 3) demonstrates that the McVittie metric accounts fully for the extra dimming of the type Ia supernovae for $z > 1$, without invoking new concepts unknown to standard physics, such as hypothetical dark energy. The only viable way of linking dark energy to standard physics is by interpreting it as the vacuum energy of space, known from particle physics experiments. But there is a huge (about 120 orders of magnitude) discrepancy between the dark energy based on the cosmological constant devised from the extra dimming of the type-Ia supernovae and the vacuum energy known from particle physics experiments. This discrepancy cannot be resolved for $\Lambda$CDM, but there is no such discrepancy in the model based on elliptical space with McVittie metric because the lambda-parameter of the latter model is interpreted as curvature $\lambda = R^{-2}$ and not as vacuum energy.



### 4.2. Dark Matter

As for dark matter, the observational evidence of its existence (e.g. the galaxy rotation curves) is as challenging for the ΛCDM as for any other alternative model. So far, all the attempts to find any experimental evidence for dark matter particles have failed, while an alternative interpretation of galaxy rotation curves (the Modified Newtonian Dynamics theory, or MOND) continues to challenge ΛCDM. The discussed McVittie metric-based model is closer to MOND in dealing with the dark matter issue (but this is a subject to be discussed in a separate paper).

As for the other observational facts, it is almost universally forgotten that, in the past, static cosmological models were not only supported by the same observational phenomena as ΛCDM, but they predicted such phenomena existed. In the literature, there are plenty of papers discussing how static-universe models solve challenging observational facts in alternative ways to ΛCDM. So, there is no need to review here these topics in detail. Below, I shall only briefly outline the most important of them.

### 4.3. Cosmological Redshift and the Cosmic Background

First of all, the cosmological redshift phenomenon was predicted by de Sitter for a static cosmological model well before the appearance of any dynamical cosmological model. Moreover, de Sitter warned in his 1917 paper that the lines of spectra systematically displaced towards the red might give rise to a spurious positive radial velocity interpretation [2]. In 1923, Eddington repeated this warning by writing that: in de Sitter's theory, there is the general displacement of spectral lines to the red in distant objects due to the slowing down of atomic vibrations which would be erroneously interpreted as a motion of recession [60].

Then, in 1926, Eddington predicted a thermalised background radiation to exist with $T = 3$ K for a static-Universe model [61]. Later, in 1937, a similar prediction with respect to the CMBR temperature $T = 2.8$ K within a static Universe framework was proposed by W. Nernst [62]. Only much later, in 1953, G. Gamow made his prediction with respect to the CMBR and its temperature $T = 7$ K for the expanding-Universe model [63]. Several other authors, e.g., [64,65], were exploring CMBR properties in the 1990s and 2000s within the framework of a static universe model. The possibility of a local origin of the CMBR was already excluded by measurements of excitation lines in absorption features of quasar spectra [66], and by measuring the imprint of galaxy clusters on the CMBR via the Sunyaev-Zeldovich effect [67].

It turns out that the model discussed here unintentionally gives yet another alternative explanation of the CMBR. As was mentioned in §2.2, that all sources belonging to a large local volume of elliptical space around an arbitrary observer are visible as uniform light emitted from within a very distant layer of space adjacent to the collective wormhole far-horizon. This possibility is in tune with de Sitter's comment about seeing "the back side of the Sun at the point of the heavens opposite to the Sun" in elliptical space [2].

For a local observer, this emission from a volume layer near the remote horizon is correspondingly redshifted due to the Schwarzschild metric. So, the energy (temperature) of this emission is scaled in accordance with the gravitational redshift formula $T(z) = T_0(1 + z)$, which is the same for the accelerated expansion of the Universe. Here $T_0$ is the highly redshifted temperature of photons produced by sources within the local sphere of matter surrounding the observer but coming to the observer from a remote layer of space adjacent to the far-horizon (in Figure 2, the local sphere of matter surrounding the observer is schematically indicated by the large-triangle pattern around the centre of the Poincaré disk, and the remote layer of space adjacent to the far-horizon is shown as the pattern of tiny triangles adjacent to the large dashed circle, the latter denoting the far-horizon location). This is a topic for discussion in a separate paper.

### 4.4. Abundances of Light Elements

The abundances of light elements in a static universe were explained by G.R. Burbidge and F. Hoyle [68,69], R. Salvaterra and A. Ferrara [70] and others, although there are some

unresolved issues for static universe models. For example, according to the standard cosmological model, deuterium ($^2$H) was created exclusively during the Big-Bang nucleosynthesis stage, after which it cannot be produced, and can only be destroyed in stars [71]. Therefore, its observed abundance is gradually diminishing. Similarly, lithium ($^7$Li) is also regarded as having been produced during the Big-Bang nucleosynthesis. However, observations suggest its continuous enrichment due to cosmic-ray spallation [72].

There are other elements (e.g., boron) that cannot be produced in stars. But it is possible to explain their existence by the same cosmic-ray spallation or fusion reactions [73]. There are numerous studies of this process in the literature, e.g., [74–76], opening an alternative understanding of light element formation, which can be used by static cosmological models [69]. In fact, the same mechanism can also explain the production of $^2$H, and replenishment of $^1$H burned in nuclear reactions in stars. Since the energies of cosmic rays can be as high as $10^{19}$ eV, they can produce spallation fragments even from $^4$He [77]. Highly energetic neutrinos can also spallate $^4$He [78]. Reactions of this kind are regularly observed in laboratory experiments [79].

Other alternatives to the Big-Bang nucleosynthesis include the synthesis of light elements in massive objects within the central regions of galaxies [68] and in extreme processes involving neutron stars [80,81].

### 4.5. Cosmic Structure

According to the classical scenarios, structure formation in static universe models occurs due to the mechanism of gravitational instability. Initial fluctuations in the homogeneous gas of primordial hydrogen grow exponentially into large-scale structures [82,83]. In the early years of cosmology, these scenarios agreed with observations. But, when Eddington showed in 1930 that Einstein's static model of the Universe was unstable [84], static models fell out of fashion.

Only fourty yeas later, Eddington's verdict was overturned by N. Rosen [85], who rehabilitated the Einstein static model and proved its stability. This reopened the possibility for solving the problem of structure formation in static cosmological models.

In these models, matter clumping was found to be fractal [21,22], which is confirmed by statistical studies of CMBR maps [20] and of the matter distribution in the Universe [17–19]. By contrast, according to the ΛCDM scenario, matter distribution cannot be fractal. The origin of the largest scale structures in the universe (the filaments and wall-like super-clusters, with huge voids between them), is not yet entirely understood from the point of view of ΛCDM.

### 4.6. Angular Sizes

Static and dynamic cosmological models predict different angular sizes on the sky of remote objects whose linear sizes are known (standard rulers). For example, in an expanding-universe model, the angular size of a remote galaxy varies in such a way that as the galaxy is moved from lower to higher redshifts its angular size decreases at first to a minimum, then increases. This theoretical prediction of the standard cosmological model with $\Omega_\Lambda = 0.7$ is illustrated by the black solid curve in Figure 6. For an empty universe with $\Omega_\Lambda = 1$, the angular size of a standard ruler always decreases (the black dotted curve).

Galaxies are not the standard rulers—their sizes vary significantly. But one can check how most of them appear at different redshifts. The theoretical curves plotted in Figure 6 correspond to a medium-size galaxy of 15 kpc. The low-redshift galaxy sizes vary about this value. Galaxies of the same size at redshift $z \sim 1$ appear to have an average angular diameter of $D \sim 1$ arcsec. This is slightly smaller than is predicted by the ΛCDM-model for a 15 kpc galaxy. So, one can conclude that galaxies evolve in time and grow by merging with other galaxies.

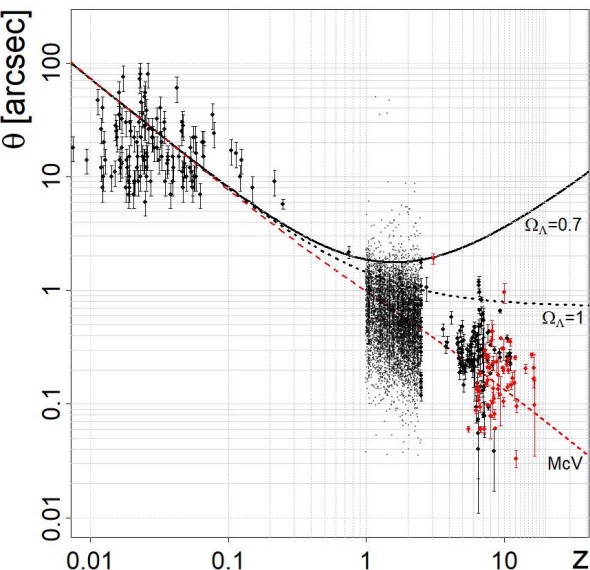

**Figure 6.** Angular sizes $\theta$ of galaxies with different redshifts. The red points indicate the data obtained by various authors from the very high-redshift observations made by the JWST during it's first year in space [86–95]. The black points indicate the data obtained before JWST [96–105]. The theoretical curves are calculated for a medium-size galaxy of 15 kpc using the standard ΛCDM model (plain black curve) with the parameters ($\Omega_\Lambda = 0.7$, $\Omega_m = 0.3$, $k = 0$). For comparison, the black dotted curve is for an empty universe ($\Omega_\Lambda = 1$, $\Omega_m = 0$). The dashed red curve (McV) shows the galaxy angular sizes as expected by the model with the McVittie metric.

At higher redshifts, $5 \lesssim z \lesssim 13$, the observed angular sizes of galaxies continue to decrease (the red points in this plot correspond to the JWST observations). They match neither the ΛCDM, nor the empty universe model's predictions for $D = 15$ kpc. Accordingly, the calculated physical sizes of these galaxies at $z \approx 10$ turn out to be very small.

At first glance, this result seems sensible, because the prediction of the Standard Cosmological Model is exactly this progression in size, from being small at large redshifts, when the Universe was very young, to large sizes at smaller redshifts. But there is a problem here: small galaxies at the initial stages of their evolution are supposed to be irregular and to have small masses.

In contradiction to this, the high-redshift galaxies are found by the JWST to be well-evolved, symmetrical and having sometimes disks and bulges. Moreover, their masses turn out to be very large, similar to the masses of nearby galaxies. In addition, such galaxies must have evolved within the very short time available since the beginning of the Universe, i.e., a few hundred million years assuming the ΛCDM Big Bang model. Moreover, the chemical composition of these galaxies and the dust in them is the same as in nearby galaxies.

Imagine a well-evolved galaxy, similar in shape, mass and chemical composition to our Milky Way, but being just a 1/10 th of the Milky Way's size. This looks very unnatural. Now, if one looks at the predictions for the sizes of the $z > 10$ galaxies made by the model with the McVittie metric discussed here (the red dashed curve in Figure 6) one finds that these galaxies are actually pretty normal, by being ~15 kPc in their sizes or more. The model discussed here provides plenty of time for these galaxies to evolve (two orders of magnitude more than ΛCDM).

## 5. Conclusions

As we have seen, the McV and ΛCDM models are equivalent for the redshift range $0 < z < 2.3$: their goodness-of-fit criteria ($\chi^2$) are identical. But for larger redshifts, these two models diverge (see the right panel of Figure 3). According to this scenario, if newly-discovered high-redshift supernova appear dimmer than what is predicted for them by the

ΛCDM model, then the McVittie metric of elliptical space would need to be considered more seriously.

**Funding:** This research received no external funding.

**Data Availability Statement:** The data used for preparing this manuscript are available at https://pantheonplussh0es.github.io (accessed on 27 March 2024).

**Acknowledgments:** This research has made use of the following archive: The *Pantheon+* Type Ia supernova distance moduli from [14,15] I am also acknowledging the use of the cosmoFns software package version 1.1-1 developed by Andrew Harris and available at the following web page: github.com/cran/cosmoFns (accessed on 27 March 2024). I would like to thank Paul Kuin, Leslie Morrison, Alice Breeveld, Victor Tostykh and Mat Page for useful discussions on the matters in this paper.

**Conflicts of Interest:** The author declares no conflict of interest.

## Abbreviations

The following abbreviations are used in this manuscript:

| | |
|---|---|
| CMBR | Cosmic Microwave Background Radiation |
| dS | de Sitter (metric) |
| FLRW | Friedmann-Lemaitre-Robertson-Walker (metric) |
| JWST | James Webb Space Telescope |
| ΛCDM | Lambda-Cold-Dark-Matter (cosmological model) |
| McV | McVittie (metric) |
| MOND | Modified Newtonian Dynamics (theory) |
| SN | supernova |
| WMAP | Wilkinson Microwave Anisotropy Probe. |

## Notes

[1]    $a(t) \propto H_0 t$; $H_0$ being the Hubble constant.

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
