# Peer review of "Elliptical Space with the McVittie Metrics"

_universe, doi:10.3390/universe10040165_

Round 1

Reviewer 1 Report

Comments and Suggestions for Authors

The topological identification of the antipodal points in elliptical space, which is its primary characteristic, may be crucial to comprehending the nature of the cosmological redshift. The computed distance moduli of Type-Ia supernovae, which are referred to as “standard candles” in cosmology, are compared by the author with the observational data that is available in the “Pantheon+” collection. The mathematical (topological) structure of elliptical space can be physically interpreted through the use of physical connections known as “wormholes,” or Einstein-Rosen bridges. These structures embedded in dynamic (growing) spacetime have a Schwarzschild metric that matches McVittie's solution to Einstein's field equations. In this metric, the time-dependent scale factor of the Friedmann-Lemaitre-Robertson-Walker metric is combined with gravitational redshift to determine the cosmic redshift of spectral lines from distant sources. Since the majority of the cosmological redshift is gravitational by nature, the constraint based on the accurate “Pantheon+” observational data yields a substantially lower expansion rate of the universe than is currently expected by modern cosmology. In the model under discussion, the estimated age of the universe is 1.48 × 1012 years, over two orders of magnitude older than the age derived from the standard cosmological model assumptions. In this scenario, the McVittie measure of elliptical space would need to be taken more seriously if recently discovered high-redshift supernovas appear dimmer than what is anticipated for them by the ΛCDM model.

I recommend publishing the article in its present form.

Author Response

Although the Reviewer does not suggest changes, I have added an extra plot at the end of the Discussion section, providing the theoretical angular-diameter distance calculation within the discussed model, which can be compared with the recent observational data from the JWST mission. This comparison shows correspondence of observational data to the predictions of the discussed model.

Reviewer 2 Report

Comments and Suggestions for Authors

The paper proposes an elliptical-space (quasi-static) model to impart another interpretation for the redshift luminosity-distance relation for the Type Ia supernovae using the Pantheon+ catalog to assess the model's constraints. The results are consistent with the predictions of the standard cosmological model (LCDM).

The interpretation of the antipodal points as a wormhole connection is not satisfactory and, frankly, quite confusing. Supposedly, it is a topological feature of the metric, so it should be stable when it is well-known that wormholes are unstable. Does this mean the model itself is unstable?

Furthermore, the results are weak. The author argues it is “as successful as LCDM” (lines 242) concerning the data fitting from Pantheon+, and argues that it could explain some features attributed to dark matter and dark energy. However, these same observations led to several tensions in the LCDM model (like Hubble and sigma_8 tensions) that are not addressed in the paper despite presenting a new prediction for the age of the universe.

Although the methods and results are presented clearly, with a proposition of a new model with measurable predictions, the paper does not have the impact and relevance the journal aims for.   

I advise against its publication.

Author Response

------------------------
Reviewer Comment 2.1.
------------------------
The interpretation of the antipodal points as a wormhole connection is not satisfactory and,  frankly, quite confusing. Supposedly, it is a topological feature of the metric, so it should  be stable when it is well-known that wormholes are unstable. Does this mean the model itself is unstable?
-------------------
Response 2.1:
-------------------
The answer to this question is "No". The model is stable, as it is checked against the observational data that cover a few billion years. By saying that "it is well-known that wormholes are unstable" the reviewer probably was referring to the 1962-paper by R.W. Fuller and K.S Thorne, who discussed the stability of wormholes traversed by matter particles or photons. This is not applied to the microscopic wormholes discussed in my manuscript. Microscopic wormholes are not traversable,  so they cannot collapse in the manner described by Fuller and Thorne.  I have added a new paragraph to Section 2.1, commenting on the wormhole stability issue.

------------------------
Reviewer Comment 2.2.
------------------------      
Furthermore, the results are weak. The author argues it is “as successful as LCDM” (lines 242)  concerning the data fitting from Pantheon+, and argues that it could explain some features  attributed to dark matter and dark energy. However, these same observations led to several  tensions in the LCDM model (like Hubble and sigma_8 tensions) that are not addressed in the  paper despite presenting a new prediction for the age of the universe.
-------------------
Response 2.2:
-------------------
It looks like the reviewer did not notice that the mentioned issues related to the LCDM-model tensions have been already highlighted in the Introduction section of the reviewed manuscript (page 3, lines 83-91). References to the previously published papers by the author, discussing these tensions in detail were provided. The so-called tensions are model-dependent. Therefore, they belong to the processing schemes of the other models, and they  are outside of the scope of the model under discussion here.  Within the discussed framework neither dark energy exists, not the related model-dependent issues. 

------------------------
Comment 2.3:
------------------------  
Although the methods and results are presented clearly, with a proposition of a new model  with measurable predictions, the paper does not have the impact and relevance the journal aims for.   
-------------------
Response 2.3:
-------------------
As the discussed model, indeed, makes measurable predictions (e.g. with respect to the future discoveries of type-Ia supernovae), the reviewer's opposition to allow these predictions to be public looks tremulous.

Reviewer 3 Report

Comments and Suggestions for Authors

The paper deals with some features of McVittie metric   and investigates the redshift-luminosity distance for type Ia-supernovae. In my opinion, the paper deserves publication. Since it deals with elliptic spaces, I would like that the author quotes the analysis contained in the paper 

"Discontinuous Normals in Non-Euclidean Geometries and

Two-Dimensional Gravity" (E. Battista and G. Esposito)

which exploits concepts analyzed in the paper 

"What is a reduced boundary in general relativity"'

(E. Battista and G. Esposito)

In any case, the paper, with or without the suggested references, deserves publication. The decision is up to the author. 

Comments on the Quality of English Language

some typos occur in the paper and should be fixed.

Author Response

------------------------
Reviewer Comment
------------------------ 
Since it deals with elliptic spaces,  I would like that the author quotes the analysis contained in the paper "Discontinuous Normals in Non-Euclidean Geometries and Two-Dimensional Gravity" (E. Battista and G. Esposito) which exploits concepts analyzed in the paper "What is a reduced boundary in general relativity"' (E. Battista and G. Esposito)

-------------------
Response:
-------------------
Indeed, the suggested works deal with elliptic geometry. But also they  discuss a scheme for constructing a topological boundary around a selected point of space in terms of algebraic topology, which is the subject of Section 2.2 of the reviewed manuscript. So, I have quoted these publications at the end of Section 2.2.  

Reviewer 4 Report

Comments and Suggestions for Authors

The perspective of explaining the extra redshift observed in the type Ia supernovae without invoking the dark energy concept is appealing.

Nevertheless, among others there are some issues that need some clarification:

-How  the porcentual matter distribution in the universe changes in this model?

-Is there any expression for H in terms of the redshift? To complement Fig. 3, it would be interesting to see the log plot of the luminosity distance Hdvs redshift compared to LCDM.

-Can give more details about the age of the universe using the radial geodesics for light traveling trough them, using the metric (23)?

-Check the writing of Eq. (20).

Author Response

----------------------------
Comment 1.
----------------------------
-How  the percentual matter distribution in the universe changes in this model?
-------------------
Response 1:
-------------------
To answer this question, I have calculated the matter-to-radiation density ratio for the period of time from the end of the radiation-dominated era to the present epoch and plotted the result in Figure 5 of the Discussion section. It turns out that, due to the  smallness of the Hubble-like constant obtained in the discussed model, it takes about an order of magnitude larger time than the age of the Universe to achieve the present-day  matter-to-radiation density ratio. That is, the age of the Universe estimated by using  the Type-Ia supernova observational data does not agree with the time needed to obtain the observed percentage of matter if using Friedmann model. The most likely reason for this inconsistency is that the derivation of the matter-to-radiation density ratio evolution based on the Friedmann equations  is not sufficient for the discussed model. Therefore, further work in this direction is needed. I have added a corresponding comment below Figure 5. 

----------------------------
Comment 2:
---------------------------
-Is there any expression for H in terms of the redshift? To complement Fig. 3, 
it would be interesting to see the log plot of the luminosity distance HdL vs redshift  compared to LCDM.
-------------------
Response 2:
------------------
From expression (20) [which is now  (21) in the revised manuscript],  it follows that the parameter H is a constant. There is no dependence on redshift or time. To show this, I have added the corresponding formula (32) at the bottom of Section 3. 

To compare the difference between the discussed model and the LCDM model, 
I have plotted the luminosity distances for both models in a new Figure 4 (left panel). Additionally, on the right panel of Figure 4 I have plotted the look-back time curves for both models. These curves approximately indicate the age of the Universe for the case of the Hubble constant 73 (km/s)/Mpc and for the case of the Hubble-like constant 0.662 (km/s)/Mpc of the discussed model. 

Also I thought that, besides the luminosity distance, it would be instructive to calculate the angular-diameter distance. Thus, I have added yet another plot, Figure 6, with the angular-diameter as a function of redshift for the discussed model and for the LCDM model. These curves are comparable  with observational data in the form of angular diameters from various galaxy catalogues,  including those obtained recently from the JWST telescope. To describe this plot I have added a new subsection (Angular sizes) to the Discussion section.

----------------------------
Comment 3:
---------------------------
-Can give more details about the age of the universe using the radial geodesics 
for light traveling trough them, using the metric (23)?

-------------------
Author's response 3:
------------------
In this particular case, the radial null geodesics will always result in the infinite time as seen by the remote observer because of the Schwarzschild metric being involved.   So, the estimation of the Universe's age is based here on the inverse of the Hubble-like constant, which was converted to the frequency units, 1/s. I have added the necessary details of this calculation by splitting Section 3 into two subsections and by highlighting the calculation details by presenting 
them in the form of a few formulae, from (30) to (33). 

----------------------------
Comment 4.
---------------------------
-Check the writing of Eq. (20).
-------------------
Response 4:
------------------
It was a misprint in the latex-code for formula (20). I have corrected this formula,
which is numbered as equation (21) in the revised manuscript,  because Section 2.1 now contains one extra formula (7).

Round 2

Reviewer 2 Report

Comments and Suggestions for Authors

The new manuscript addressed the issues previously pointed out in an unsatisfactory way.

The instability of wormholes is far from a “belief” that one can dismiss so easily as to simply add that “microscopic wormholes are not transversable,” and thus conclude they are stable without even providing a reference. The reference cited by the authors on the macroscopic wormholes [KOIRAN, 2021] does not address the stability issue. It very clearly provides an interpretation for a particle on an inward trajectory crossing the bridge, while stability is more than complete geodesic.

Furthermore, moving the observer’s point-of-view from one end of the wormhole to the other requires a transversable wormhole, and the authors claimed the one they considered are NOT transversable. They are correct when they say there are studies indicating the stability of microwormholes, although they cite none. On the stability of microscopic transversable micro wormholes, I refer to [BLÁZQUEZ-SALCEDO et. al. 2021]. As clearly presented in the paper, the conditions for stability are far from trivial and do not match the author’s model.

I would like to remark that mentioning an issue is not the same as addressing it, and that a model with measurable predictions does not equate to quality science. The authors are free to make their work publicly in any preprint archive of their choice. I stand by my previous assessment that the results are weak, and I will add that the model has fundamental flaws in its construction that the authors failed to explain. Thus, I do not recommend the publication of this paper in this journal.

[BLÁZQUEZ-SALCEDO et. al. 2021] Jose Luis Blázquez-Salcedo, Christian Knoll, and Eugen Radu. Phys. Rev. Lett. 126, 101102, 2021. DOI: https://doi.org/10.1103/PhysRevLett.126.101102

Author Response

Reviewer's comment-1.  The instability of wormholes is far from a “belief” that one can dismiss so easily  as to simply add that “microscopic wormholes are not transversable,” and thus conclude  they are stable without even providing a reference. The reference cited by the authors  on the macroscopic wormholes [KOIRAN, 2021] does not address the stability issue.  It very clearly provides an interpretation for a particle on an inward trajectory  crossing the bridge, while stability is more than complete geodesic.

Response-1
I agree that the cited reference does not address the stability issue. It addresses
the issue of traversability. However, these two issues are intimately related to each other because the main point of all stability or traversability studies is penetration of photons or matter particles to the inner parts of wormholes, like in the 1962 stability by Fuller and Thorne. Moreover, in the literature, the terms ``traversable" and "non-traversable" became synonyms to the terms "macroscopic" and "microscopic". In this manuscript, it is the microscopic structure that is discussed, and not the macroscopic. I have added the corresponding comments and a few corresponding references to Section 2.1.

Reviewer's comment-2
Furthermore, moving the observer’s point-of-view from one end of the wormhole  to the other requires a transversable wormhole, and the authors claimed the one they considered are NOT transversable. They are correct when they say there are studies  indicating the stability of microwormholes, although they cite none. On the stability  of microscopic transversable micro wormholes, I refer to [BLÁZQUEZ-SALCEDO et. al. 2021].  As clearly presented in the paper, the conditions for stability are far from trivial  and do not match the author’s model.

Response-2:
This comment is wrong. It indicates that the reviewer has not understood the main point of the manuscript. Moving the observer's point-of-view from one end of the wormhole to the other DOES NOT require the wormhole to be traversable. All the photon trajectories are OUTSIDE of the wormhole's  horizons. The horizons are not crossed and cannot be crossed, as it would take infinite time
from the observer's perspective.  As in the previous comment, I can highlight that all the references related to the wormhole's traversability or stability, including  BLAZQUEZ-SALCEDO et. al. 2021], are not appropriate for this discussion because they are  related to macroscopic wormholes, whereas it is underlined a few times in the manuscript that what is discussed here is space filled in with MICROSCOPIC wormholes. 

Reviewer's comment-3:
I would like to remark that mentioning an issue is not the same as addressing it, 
and that a model with measurable predictions does not equate to quality science. 

Response-3 
The mentioned issue is outside of the scope of this manuscript, while the discussed measurable predictions have nothing to do with the question about wormhole stability or traversability. I think that the reviewer intentionally sidetracks the discussion from the main point the manuscript because this point was not understood by the Reviewer, as can be clearly seen from the reviewer's comment-2.